REGISTERED REPORT PROTOCOL

# Knowledge about research and facilitation of co-creation with children. Protocol for the article "scoping review of research about co-creation with children"

Bjarnhild Samland [ID]*, Tone Larsen, Lillian Pedersen

Western Norway University of Applied Science, Bergen, Norway

* Bjarnhild.samland@hvl.no

## Abstract

Children and young people's participation, as stipulated in the Convention on the Rights of the Child, applies to both matters that directly and indirectly affect children. Participation is in some countries recognized as a fundamental right and children's engagement seen as a valuable resource. Assisted by conceptual understanding of co-creation, children may be enabled to engage and participate in a variety of contexts. Knowledge about research on, and facilitation of, co-creation involving children is the theme of the scoping review presented by this protocol. The protocol outlines a scoping review which is to use a systematic approach to synthesize knowledge of research about co-creation with children. By systematically scoping the existing research about co-creation with children, the review will survey the available literature (evidence), identify key concepts, and uncover gaps in knowledge. The overall objective of this scoping review is to gain knowledge of research conducted about all types of co-creation with children, and to identify the gaps that future research should address. This scoping review acknowledges the existence of multiple definitions of co-creation, which vary depending on different contexts. The review will also recognize several other associated concepts, such as co-production, co-design, co-research, and co-innovation, since they are used interchangeably with or align with the understanding of co-creation being reviewed. The methodological framework outlined by the Joanna Briggs Institute (JBI) for scoping review will be used as a guide for this review. The PRISMA Extension for Scoping Reviews (PRISMA-ScR): Checklist and Explanation will be used during the process. The databases, ERIC (Education Resources Information Centre), Teacher Reference Center, Idunn, Oria, Libris, Kungliga biblioteket, ScienceDirect, ProQuest, Scopus, Academic search elite, Web of Science, Google scholar, will be searched for information on academic books and articles, in May 2024. Also grey literature will be searched for relevant academic references. There are no limitations in date of publication. Language will be limited to English, Norwegian, Swedish, and Danish. Following the selection of studies, data will be extracted and analysed. Ethical approval is not required, because only secondary data is collected. Dissemination will include peer-reviewed publications and presentations at conferences regarding public innovation, education, and children's participations contexts.

**Data Availability Statement:** All relevant data from this study will be made available together with this manuskript upon study completion.

**Funding:** This study was funded by the Norwegian Research Council, Western Norway University of Applied Sciences, and Sogndal Municipality.

**Competing interests:** The authors have declared that no competing interests exist.

# Introduction

The aim of this scoping review is to investigate research including facilitation of co-creation with children, with a focus on methodologies and methods. Through this review, we will illuminate approaches that have been employed in researching co-creation with children.

The perspective on children has undergone a transformation in recent decades. Instead of viewing children merely as *becomings*, we now recognize them as *beings* with their own rights and agency. This shift in view has been solidified through the UN Convention on the Rights of the Child, with particular emphasis on article 12 [1, 2]. This article explicitly establishes children's entitlement to express their views on matters affecting them directly, indirectly, and within the broader societal framework [1, 2]. However, a gap between the stated article and its realisation within policy and practice contexts has been found in Robinson [3]. Therefore, it seems crucial to gain insight into how children's voices and perspectives can be included. In this context, co-creation is an approach that may include children and promote their perspectives and voices.

The concept of co-creation emerged in the private sector, where service providers'collaboration with consumers to create value together, known as value co-creation, was deemed beneficial [4, 5]. Co-creation is defined in various ways depending on different disciplines (e.g., marketing, service management, public management) and the context which it is applied. However, researchers argue that in the public sector, co-creation involves collaboration between public, private and/or civil actors to create public value by sharing knowledge and resources [5–9]. Ramaswamy & Ozcan [4 p.14] define co-creation as follows:

> "Co-creation is joint creation and evolution of value *with stakeholders' individuals*, intensifies and enacted through *platforms of engagements*, virtualized and emerging from *ecosystems of capabilities* and actualized and embodied in *domains of experiences*, expanding *wealth–welfare–wellbeing*."

The elements of co-creation emphasized in this definition, *value creation*, *stakeholders' perspectives*, *and arenas for co-creation*, suggest that co-creation is a multifaceted phenomenon that can be described using various terms and definitions, such as social innovation, co-design, co-production, and co-creation itself [10]. In literature, about the public sector, the terms "co-creation" and "co-production" are often used interchangeably, representing collaborative efforts between civil society and public servants to initiate, plan, design, and implement public services [11]. However, the term co-creation is more commonly used in the context of initiating or designing services, while co-production means involving residents in the service implementation stage [5, 9]. Co-production can be understood as a process where the public organisations hold dominance and focusing on linear production. In co-creation the relationship between stakeholders appears as more interactive, equal, and dynamic. Here, value is created in the interaction that takes place within the context of the service user's wider life experience [5]. The focus of the scoping review is investigating research projects that include co-creation with children, but related concepts will be taken into consideration if the collaborative interaction, a) includes children, and b) aims to create public and private value. Another precondition is that different stakeholders engage in a collaborative process with children, both providers and users of a service or product, who aim to promote innovation and improvement. This innovative dimension may lead to terms and concepts that are related to co-creation, such as co-innovation [6, 10]. In our search we thus included the concepts of co-creation, co-innovation, co-production, co-design, and co-research.

In addition to examining the facilitation of co-creation with children, we will explore how research on co-creation has been conducted. An area of special interest will be to investigate

how children's perspectives on the co-creation process are illuminated. Recently, there has been an expansion in the field of participatory research involving children, driven by a growing emphasis on conducting research *with* children rather than just *on* them [12]. There is wide diversity in the ways participatory research with children is conducted, and there is increased attention to the ethical dimensions such as power, language and roles [12–14]. This review aims to explore research, identify, and analyse knowledge gaps, and provide guidance for future research of co-creation that supports further implementation of the Convention on the Rights of the Child in research and society at large. The review question we will explore is: *What knowledge exists about the facilitation of co-creation with children and how has the research been conducted*?

The decision to conduct a scoping review was based on a careful assessment of available research methods. While a systematic literature review was initially considered, a scoping review was deemed to be more appropriate due to the need for a comprehensive overview of a broad topic [15]. There is no universally accepted definition of scoping reviews [15]. Munn et al. [16 p.950], reached a formal consensus with the JBI Scoping Reviews Methodology Group in 2020 on the following definition of scoping reviews:

> Scoping reviews are a type of evidence synthesis that aims to systematically identify and map the breadth of evidence available on a particular topic, field, concept, or issue, often irrespective of source (i.e., primary research, reviews, non-empirical evidence) within or across particular contexts. Scoping reviews can clarify key concepts/definitions in the literature and identify key characteristics or factors related to a concept, including those related to methodological research.

The key elements of a scoping review [16] are what is required to address the objective of the review: The objective is to assess the extent of the academic literature that describes co-creation processes involving children, examine how research is conducted, and identify key characteristics or factors related to research about co-creation with children. This scoping review aims to elucidate the existing knowledge regarding children's participation in research on co-creation. It may also contribute to the knowledge development concerning facilitation of co creation involving children. It may identify and illuminate exemplary practices in various contexts, such as education, pedagogy, product and service development, cultural and social development. It serves to offer a comprehensive overview of the diverse contexts in which children engage in co-creation and might show how research methods affect co-creation processes. Additionally, the review serves to enhance theoretical and methodological frameworks within the realm of co-creation involving both children and adults.

A preliminary search of ERIC and Scopus is conducted and no current or underway systematic reviews or scoping reviews on the topic were identified. However, Williams et al. [17] conducted a scoping review to explore co-creation involving children. It is worth noting that while our research shares some common themes, their scope and context is limited to the enhancement of health-promoting physical environments within publicly accessible spaces.

## Methods

The proposed scoping review, to identify and explore the concept of co-creation with children, will be conducted in accordance with the JBI methodology for scoping reviews [18] and the study design presented in this protocol has been checked according to Peters et al. [19 p. 956] "Checklist of the Preferred Reporting Items for Systematic Reviews and Meta-Analyses Protocols (PRISMA-P) statement adapted for a scoping review protocol".

The execution of the evidence synthesis will be conducted in accordance with the "Preferred Reporting Items for Systematic Reviews and Meta-analyses extension for scoping review (PRISMA-ScR) Checklist" [20]. The use of the PRISMA-ScR checklist will serve as a quality assessment for the study and a guideline for presenting the results of the search and the study inclusion process in the final scoping review.

## Types of sources

The scoping review will include existing academic literature on research and facilitation of co-creation involving children and young people aged 0–18 that has been part of a co-creation processes without any contextual limitations. Our focus will be on literature that describes co-creation that contains partnerships and innovative elements where children actively participate. Co-creation without children as participants will be excluded. We will focus exclusively on academic articles, as research methods are crucial to this review. Gray literature or individual case reports will not be included in the synthesis. The scoping review's focus is to examine how children's perspectives are integrated into research and the potential connections between research on, and facilitation of, co-creation where they participate. Thus, we will read the abstracts of grey literature that occur in our search and scan them for references to relevant academic literature research.

To ensure a comprehensive analysis, we will consider studies employing various research designs, both qualitative and quantitative approaches, including action research. By incorporating diverse research designs, we aim to capture a broad range of perspectives and approaches in the literature pertaining to co-creation with children.

## Search strategy

The reviewers have collaborated with an academic librarian, who is an expert in developing and performing searches for systematic reviews and meta-analyses, to prepare the search strategy.

This ensures transparency and auditability of both the search methodology and its outcomes. The search strategy is not limited by study design or year of dissemination and has been peer reviewed by another information specialist using the Peer Review of Electronic Search Strategies (PRESS) checklist [21].

The search is limited to title, abstract and keywords. We have undertaken an initial limited search of ERIC (Education Resource Information Centre) and Scopus, to develop our search strategy. We identified several articles on the topic. The text words contained in the titles and abstracts of relevant articles, and the index terms used to describe the articles have been used to develop a full search strategy. Literature in English, Norwegian, Swedish, and Danish will be included. The databases ERIC, Idunn, Oria, Libris, Kungliga biblioteket, ScienceDirect, ProQuest, Scopus, Academic search elite, Web of Science and Google scholar will be searched for information. The search strategy, including all identified keywords and index terms, will be adapted for each included database. The literature search will be supplemented by manually scanning the reference lists of included studies to identify further publications linked to them. Furthermore, academic experts in co-creation with children will also be contacted to obtain information on relevant literature.

We have translated the search-words into the Scandinavian languages, due to our ability to understand them. This increases the chance to identify potentially overlooked scientific articles, especially for text searches. However, the search databases used in our search are international and use English index words (thesaurus). This means that our search strategy can retrieve articles across all languages, provided they are indexed in these databases. To ensure

**Table 1. Search strategy.**

| Aspect (key words) | Thesaurus (index terms) | Textwords |
|---|---|---|
| Co-creation | | |
| | | co W0 (creat* or produc* or research* or design* or innovat*) |
| | | cocreation* OR coproduc* OR coresearch* OR codesign* OR coinnovat* |
| Children | | |
| | | child* OR schoolchild* OR adolescen* OR boy OR boys OR boyhood OR girl OR girls OR girlhood OR youth OR youths OR teen OR teens OR teenager* OR newborn* OR puberty OR youngster* OR kid OR kids OR minors or juvenile* OR (young W0 people) OR underage* OR (under W0 age*) |
| | Young children | |
| | Children | |
| | Adolescents | |
| | Early adolescents | |
| | Youth | |
| | Preschool Children | |
| | Elementary School Students | |
| | Secondary School Students | |
| | Middle School Students | |
| | Junior High School Students | |
| | High School Students | |
| | | (0 or 1 or 2 or 3 or 4 or 5 or 6 or 7 or 8 or 9 or 10 or 11 or 12 or 13 or 14 or 15 or 16 or 17 or 18 or zero one or two or three or four or five or six or seven or eight or nine or ten or eleven or twelve or thirteen or fourteen or fifteen or sixteen or seventeen or eighteen) W0 year* W0 old* |

that the research results are as relevant as possible, we use proximity indicators "WO". Table 1 below shows the search concept that will be conducted.

## Study/source of evidence selection

Following the search, all identified citations will be collated and uploaded into *EndNote 20* and duplicates removed. After conducting a pilot, titles and abstracts will be screened by the authors for assessment against the inclusion criteria for the review.

The full text of selected citations will be assessed in detail against the inclusion criteria.

Reasons for exclusion of sources of evidence at full text that do not meet the inclusion criteria will be recorded and reported in the scoping review. The Table 2 below summarises the inclusion and exclusion criteria of this review.

Any disagreements that arise between the authors at each stage of the selection process will be resolved through discussion, or with an additional reviewer/s. To ensure uniformity in the evaluation of articles and to promote a shared understanding of the selection of sources among us, we will initiate a pilot phase. In the pilot all three authors will consider the same ten articles, according to a predetermined template for analysis and synthesis, to foster cohesion in our approach and align our assessments. We will adjust our subsequent evidence selection based on the insights gained from reflecting our pilot's outcome. To detail the study selection process, a flow diagram will be used (Fig 1).

**Table 2. Summary of inclusion and exclusion criteria.**

| Inclusion | Exclusion |
|---|---|
| 1. Academic literature utilizing both qualitative and quantitative methodologies.<br>2. Literature exploring the concept of co-creation involving children aged 0 to 18.<br>3. Academic literature that exam related concepts targeting children aged 0 to 18, encompassing elements of partnership and innovation.<br>4. Articles written in English, Danish, Swedish, and Norwegian. An exception can be made if the article's abstract is in English and appears relevant. | 1.Grey literature.<br>2. Co-creation or related concepts where children are *not* involved. |

## Data extraction

The articles will be handled using Endnote and an online software for conducting reviews, Rayyan [22]. Data will be extracted from papers included in the scoping review. The initial stages of the selection process, encompassing the following steps, will be executed by the primary author (Samland, B.):

1. Removal of duplicates

2. Exclusion of editorials, reports, opinion papers, and conference papers

3. Elimination of studies that are not conducted in or related to co-creation with children.

In instances where a study's inclusion status cannot be determined based on the abstract alone, it will be included for further consideration. The second and third authors will

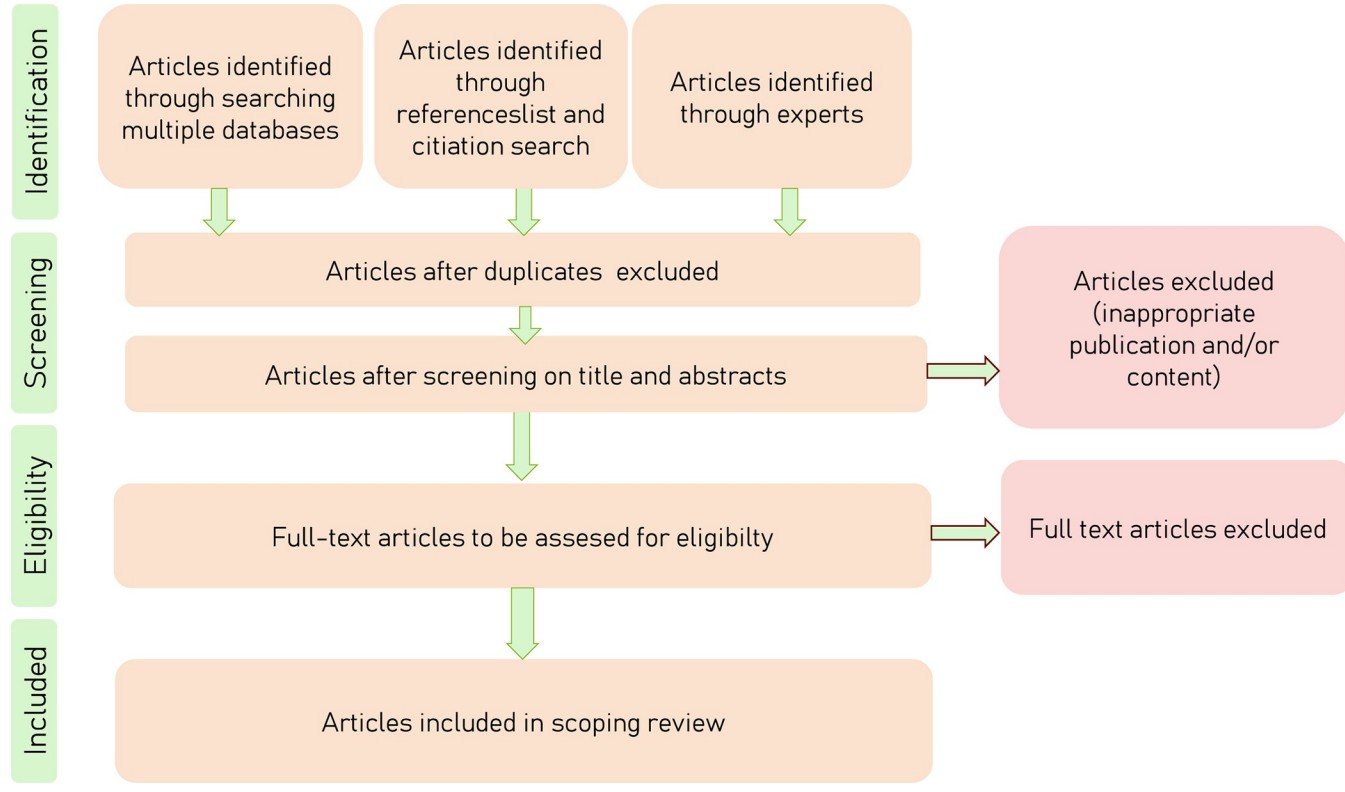

**Fig 1. Flow diagram.**

independently review 20% of the records randomly. Upon completion of this, the first author will read all the included articles to verify their adherence to the eligibility criteria. Additionally, the second and third authors will each read 20% of the studies randomly selected. Subsequently, all three authors will read through all included articles, extracting data from each included study. The selection criteria outlined in this protocol are designed to identify crucial information for describing the scope, foster theoretical development, and reveal knowledge gaps, to offer recommendations to guide future research.

The data extracted will include specific details about:

1. Author(s).

2. Year of publication.

3. Country of origin.

4. Context.

5. Aims/purpose of co-creation.

6. Concept. Process of co-creation.

7. Participants (children, pupils' representatives, co- researchers, service-users, designers, teachers, leaders, politicians, volunteers, services, organisations)

8. Methodology / methods. Facilitation and research methods.

9. Outcome (for; children, service-users, services, organisations) (e.g. how was outcome measured).

10. Key findings that relate to the scoping review question/s.

Authors must establish a shared understanding of the data extraction process. We will conduct a pilot, as with the selection of sources. During the pilot phase, all authors will individually extract data from the same three articles. Subsequently, we will meet to discuss our findings, reflect on our results, and adjust our understanding, and appraise further extraction. Throughout the data extraction process, the initial draft of the data extraction tool will be subject to revisions and adjustments as required. All modifications made will be thoroughly documented within the scoping review. Any disagreements arising between the authors will be resolved through discussion, or if necessary, with the input of an additional reviewer/s. If appropriate, authors of papers will be contacted to request missing or additional data, where required.

## Presentation of results

We will use Excel software to create an overview of the detected patterns in the articles and the extracted data. This tool enables us to synthesize and summarize the existing literature on co-creation with children clearly. Our findings will be presented using a combination of textual descriptions, tabular data, and graphical representations to ensure a useful presentation of our findings [20, 23]. All data underlying the findings, will be fully available without restriction, as part of this manuscript, at the time of publication.

## Limitations

We are aware of different limitations conducting the scoping review. Linguistic limitations exist. Although we have the advantage of mastering Scandinavian and English languages, language barriers will inevitably limit our access to all relevant literature. However, if English

abstract provides sufficient information, we may include these studies with the help of today's technology. Our preliminary test searches suggest that language will not necessarily pose a significant challenge, thanks to our broad search strategy. However, this broad search strategy leads us to the next limitation: it may reduce the precision in relation to the review's purpose. This was indicated by our test searches, but the test search also showed the necessity of a comprehensive approach to cover related concepts. Finally, it should be mentioned that the scoping review will not advocate for specific methods for exploring or implementing co-creation with children. Future work may address this. Hopefully, this scoping review may serve as a basis for systematic reviews and additional research into children's participation in co-creation and related areas.

## Conclusion

This protocol serves as a guideline for investigating the contributions of research on children's participation in co-creation processes, and how children's perspectives are highlighted. By synthesizing knowledge about research and facilitation of co-creation with children, we hope to influence children's involvement in future research as well as the facilitation of co-creation across various sectors of society. As authors, we commit to maintaining ongoing dialogue throughout the process and making necessary adjustments along the way. We will conduct continuous quality assessments, with the goal of publishing the review in a suitable peer-reviewed journal. This will make the knowledge available so it can influence, and hopefully, strengthen children's participation in different parts of the society.

## Supporting information

**S1 Checklist. Checklist of the Preferred Reporting Items for Systematic Reviews and Meta-Analyses Protocols (PRISMA-P) statement adapted for a scoping review protocol.** (DOCX)

## Acknowledgments

This protocol for a scoping review is related to the first authors´ PhD research project, "Co-creation in the public sector. Children as co-researchers".

## Author Contributions

**Conceptualization:** Bjarnhild Samland, Tone Larsen.

**Data curation:** Bjarnhild Samland.

**Formal analysis:** Bjarnhild Samland, Tone Larsen, Lillian Pedersen.

**Investigation:** Bjarnhild Samland, Tone Larsen.

**Methodology:** Bjarnhild Samland.

**Project administration:** Bjarnhild Samland.

**Supervision:** Lillian Pedersen.

**Validation:** Bjarnhild Samland, Tone Larsen.

**Visualization:** Bjarnhild Samland.

**Writing – original draft:** Bjarnhild Samland, Lillian Pedersen.

**Writing – review & editing:** Bjarnhild Samland, Tone Larsen, Lillian Pedersen.

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
