## [Decision Letter · Decision Letter 0]

18 Jan 2024

PONE-D-23-36267Knowledge about research and facilitation of co-creation with children. Protocol for the article “Scoping review of research about co-creation with children”PLOS ONE

Dear Dr. Samland,

Thank you for submitting your manuscript to PLOS ONE. After careful consideration, we feel that it has merit but does not fully meet PLOS ONE’s publication criteria as it currently stands. Therefore, we invite you to submit a revised version of the manuscript that addresses the points raised during the review process.

We look forward to receiving your revised manuscript.

Kind regards,

Sherief Ghozy, M.D.

Academic Editor

PLOS ONE

2. Please amend either the title on the online submission form (via Edit Submission) or the title in the manuscript so that they are identical.

Reviewers' comments:

Reviewer's Responses to Questions

**Comments to the Author**

1. Does the manuscript provide a valid rationale for the proposed study, with clearly identified and justified research questions?

Reviewer #1: Partly

Reviewer #2: Partly

2. Is the protocol technically sound and planned in a manner that will lead to a meaningful outcome and allow testing the stated hypotheses?

Reviewer #1: Partly

Reviewer #2: Yes

3. Is the methodology feasible and described in sufficient detail to allow the work to be replicable?

Reviewer #1: No

Reviewer #2: Yes

4. Have the authors described where all data underlying the findings will be made available when the study is complete?

Reviewer #1: No

Reviewer #2: Yes

5. Is the manuscript presented in an intelligible fashion and written in standard English?

Reviewer #1: Yes

Reviewer #2: Yes

6. Review Comments to the Author

You may also provide optional suggestions and comments to authors that they might find helpful in planning their study.

Reviewer #1: Thank you for the opportunity to review this protocol detailing a planned scoping review about co-creation with children. I have some major and minor concerns with the article, which are detailed below. My major concerns generally relate to a lack of clarity about the scope of the review and the procedures that will be employed to conduct it. In addition, there are several areas where the authors need to report information that is not currently present, such as how results will be synthesized, whether critical appraisal will be conducted, etc. It is my opinion that these issues need to be adequately addressed in order for the protocol to be publishable and useful to readers.

Introduction:

1. The research question and the scope of the review was not entirely clear to me. Will the authors be looking at articles that report on research about co-creation research? Or will they be looking at articles that use co-creation as a methodology in their research to determine how co-creation has been used?

Also, it was not clear to me whether the authors will be limiting their eligibility to studies that specifically discuss co-creation in the context of research or if they will also include co-creation with children for other purposes (e.g., to develop policies).

The authors also suggest that they may expand the search to examine co-creation in general, and not just specific to children.

I think the authors need to revise the introduction and research question to make it clear what exactly their research question and scope of their review is, and under which conditions they will expand their criteria. Perhaps the authors need to conduct some preliminary searches to understand whether they will need to expand their search or not.

2. I think the background section of the introduction could also be revised for clarity. The authors seem to jump around in their explanation of co-creation. A key line seems to be “Recently, there has been increased attention to doing research with children, instead of just on them”, however this line appears late in the introduction and is not elaborated on. I think readers may benefit from a fuller explanation of how co-creation with children relates to other participatory research approaches that readers are more likely to be familiar with (e.g., community-based participatory action research). In my opinion, this will help readers to understand the topic better and how it fits in with the larger universe of relevant research.

Methods:

1. I was confused about the types of sources you will include. Will you include grey literature or will you only use grey literature in order to find additional academic articles?

2. Some strengths of the methodology: Authors have consulted an academic librarian, had their search peer reviewed, and will be searching several databases in multiple languages.

3. The table has no title. It is also set up in a way that is not very clear. I’m not sure that the middle column is needed or what it is adding.

4. Some text in the Methods section is in different sizes/colours.

5. I have a few comments on the search with the caveat that database searching is not my expertise. I’m not clear on what the ‘W0’ search term is. Some words seem like they could be truncated instead of repeated (e.g., boy OR boys OR boyhood perhaps could be captured with boy*?). Will these searches be translated directly into the other languages or will any language-specific modifications be made to the terms?

6. Not clear how the “search and study inclusion process will be reported in full in the scoping review and presented in the PRISMA checklist.” PRISMA checklist is more for making sure you report all components of your review in the resulting article.

7. Authors report “in instances where a study’s inclusion status cannot be determined based on the abstract alone, it will not be included for further consideration.” It seems if the eligibility cannot be confirmed from the abstract, the full text should be consulted for eligibility.

8. Will all authors conduct the data extraction? Will there be any verification of the data extraction or piloting to make sure that all authors are extracting it in the same way?

9. Though the authors describe inclusion/exclusion criteria throughout the paper, I think there should be a section reporting exclusion and inclusion criteria very clearly given this is a very important aspect of the review process.

10. Will any quality appraisal be conducted?

11. How will the results be synthesized?

12. Why were these data fields selected for extraction? What value are the authors hoping this review will add to the literature? Some discussion of this would be useful to conclude the article.

13. Figure 1 is very blurry and hard to read. Colouring should be changed. I don’t think the n = should be included. Some spelling errors in this figure.

Other minor comments:

1. This manuscript has some minor grammar and spelling errors that should be fixed (e.g., the research question states “Wath knowledge…”).

2. The author report funding but their PRISMA checklist says funding is “not applicable.”

Reviewer #2: Thank you for letting me read this interesting and important paper. There is a huge need for collect and compile research on this topic to guide further research and policy. I have some minor comments for the authors´ consideration:

Abstract: lack of some background - why is there an need for this scoping review

Introduction: there is a need for some more info on why there is a need for this study - it address the concepts in a good way, but not so much on the field of co-creation with children and young people

Method: there is a need for more detailed info on inclusion and exclusion criteria, and also how they will develop a screening guide

7. PLOS authors have the option to publish the peer review history of their article (what does this mean?). If published, this will include your full peer review and any attached files.

Reviewer #1: No

Reviewer #2: **Yes: **Ottar Ness

---

## [Author Response · Author response to Decision Letter 0]

5 Jul 2024

Dear Reviewers, 

Thank you for your comments on our manuscript “Knowledge about research and facilitation of co-creation with children (for the article “Scoping review of research about co-creation with children”) 

This has been most helpful for the revision and upgrading of the manuscript. We have substantially revised our manuscript based on the reviewers’ comments and are resubmitting it for further consideration, as requested. The manuscript has been revised by using red writing. Please find below the point-by-point responses to the comments to the authors from the reviewers: 

Abstract: 

Reviewer 2: 

Abstract: lack of some background - why is there a need for this scoping review

Answer: Thanks for your valuable comment to our abstract and we hope we have more clearly expressed the need for this study. 

Introduction

Reviewer 1. 

1. The research question and the scope of the review was not entirely clear to me. Will the authors be looking at articles that report on research about co-creation research? Or will they be looking at articles that use co-creation as a methodology in their research to determine how co-creation has been used?

Also, it was not clear to me whether the authors will be limiting their eligibility to studies that specifically discuss co-creation in the context of research or if they will also include co-creation with children for other purposes (e.g., to develop policies).

The authors also suggest that they may expand the search to examine co-creation in general, and not just specific to children.

I think the authors need to revise the introduction and research question to make it clear what exactly their research question and scope of their review is, and under which conditions they will expand their criteria. Perhaps the authors need to conduct some preliminary searches to understand whether they will need to expand their search or not.

Answer: 

Thanks for giving us the opportunity to correct information that may lead to misunderstandings. We have revised the introduction and made the research question and the scope of the review clearer e.g. that we will only explore co-creation with children and both the research methods, and the facilitation are in our interest. As a part of the process to develop the search strategy, we conducted pilot searchers in different databases, but sure there will be need to adjust our search strategy as we go along. 

2. I think the background section of the introduction could also be revised for clarity. The authors seem to jump around in their explanation of co-creation. A key line seems to be “Recently, there has been increased attention to doing research with children, instead of just on them”, however this line appears late in the introduction and is not elaborated on. I think readers may benefit from a fuller explanation of how co-creation with children relates to other participatory research approaches that readers are more likely to be familiar with (e.g., community-based participatory action research). In my opinion, this will help readers to understand the topic better and how it fits in with the larger universe of relevant research.

Answer: We assume that there are several different research approaches used to explore co-creation with children, not limited to participatory research. A research method can incorporate elements of co-creation and in the participatory paradigm co-creation is sentral. We have taken note of your feedback and hopefully are the concept of co-creation and the topic of the review better explained in the new version of our manuscript.

Reviewer 2 

There is a need for some more info on why there is a need for this study - it addresses the concepts in a good way, but not so much on the field of co-creation with children and young people

Answer: Thanks for bringing this to our attention. We have tried to express this more clearly by pointing out that there is a need to gain insight into how children’s voices and perspective can be included. A significant motivation for this is the mismatch between the principles outlined in Article 12 of the Children's Convention and their actual implementation within policy and practice contexts. As far as we have experienced there are also absence of theoretical knowledge concerning co-creation with children specifically. 

Method

1. I was confused about the types of sources you will include. Will you include grey literature, or will you only use grey literature in order to find additional academic articles?

Answer: We appreciate you making us aware of this. We have made it clear that grey literature and reports only will be scanned to find relevant information and references to academic literature. 

2. Some strengths of the methodology: Authors have consulted an academic librarian, had their search peer reviewed, and will be searching several databases in multiple languages.

Answer: Thanks for your comment. 

3. The table has no title. It is also set up in a way that is not very clear. I’m not sure that the middle column is needed or what it is adding.

Answer: The table has got a title in the new version. Thanks for making us aware of this and for sharing your thoughts about the middle column so we can explain why it is so. To be sure about the middle column we have consulted our searching expert. The mid-section of the collum is the index word, thesaurus, which articles are tagged with when they are included in a database. We need both the text words and the thesaurus since these words not necessarily appear in the texts and vice versa. Therefore, we will keep the table as it is, but have added some explanations in the table and in the text above the table. 

4. Some text in the Methods section is in different sizes/colours.

Answer: Size and colour of text in the method section has been corrected. 

5. I have a few comments on the search with the caveat that database searching is not my expertise. I’m not clear on what the ‘W0’ search term is. Some words seem like they could be truncated instead of repeated (e.g., boy OR boys OR boyhood perhaps could be captured with boy*?). Will these searches be translated directly into the other languages, or will any language-specific modifications be made to the terms?

Answer: 

 “WO” are proximity indicators that ensure that the search result is as relevant as possible. An example is "co-creation". Some search databases will, without WO, give hits on articles that have "co" and "create" in the text even if the words are not connected to each other. By using proximity indicators, we thus avoid irrelevant articles. Regarding the truncation, we do not truncate the words the reviewer mentions to avoid irrelevant matches. If we truncate "boy", we will get hits on many words that are not relevant to our search, such as "boyfriend" and "boysenberries".

We have translated the search into Norwegian, Swedish, and Danish. This leads to some language-specific changes and differences, such as the fact that Norwegian does not have words that directly correspond to "boyhood" and "girlhood". Nevertheless, it is important to point out that the search databases we use are international and we apply English subject words (thesaurus) which will mean that we capture articles in all languages if they are available in the databases.

6. Not clear how the “search and study inclusion process will be reported in full in the scoping review and presented in the PRISMA checklist.” PRISMA checklist is more for making sure you report all components of your review in the resulting article.

Answer: Thanks for your notification. This was expressed wrong. This is made clearer in the text by stressing that the PRISMA checklist are for the purpose of quality appraisal the study and not a tool for presenting the results. 

7. Authors report “in instances where a study’s inclusion status cannot be determined based on the abstract alone, it will not be included for further consideration.” It seems if the eligibility cannot be confirmed from the abstract, the full text should be consulted for eligibility.

Answer: This is a mistake and we have corrected it to “In instances where a study's inclusion status cannot be determined based on the abstract alone, it will be included for further consideration.” 

8. Will all authors conduct the data extraction? Will there be any verification of the data extraction or piloting to make sure that all authors are extracting it in the same way?

Answer: All three authors will do data extraction. We will conduct a pilot and have collaborative meetings during the process, both for source selection and data extraction. This is now explained in the manuscript. 

9. Though the authors describe inclusion/exclusion criteria throughout the paper, I think there should be a section reporting exclusion and inclusion criteria very clearly given this is a very important aspect of the review process.

Answer: Thanks for your suggestion. We added a table, table 2, that summarize the exclusion and inclusion criteria. 

10. Will any quality appraisal be conducted?

Answer: We think that the review now clarifies several measures that may ensure the quality of the work. In the manuscript we have mentioned that the search is developed, tested, and carried out by a search specialist and quality assured with another specialist, we are going to do pilots both on source selection and data extraction and use of PRISMA checklist. 

11. How will the results be synthesized?

Answer: This was a useful question. We will use the PRISMA-ScR checklist as a guideline for this evidence synthesis, and we have added a chapter called “Presentation of results” where we describe briefly how we are going to synthesize the results. 

12. Why were these data fields selected for extraction? What value are the authors hoping this review will add to the literature? Some discussion of this would be useful to conclude the article.

Answer: The selection of data fields selected for data extraction is now more clearly explained in the chapter about data extraction. We have made the purpose of this review more clearly both in the abstract and in the introduction chapter. We have also added a chapter “Conclusion” that will sum up this point. 

13. Figure 1 is very blurry and hard to read. Colouring should be changed. I don’t think the n = should be included. Some spelling errors in this figure.

Answer: The figure has been improved in accordance with your useful feedback. 

Reviewer 2 Method: there is a need for more detailed info on inclusion and exclusion criteria, and also how they will develop a screening guide.

Answer: Hopefully, we have obtained more detailed information about inclusion and exclusion criteria by better explaining and creating a table summarizing the criteria. Concerning a screening guide, we have outlined a pilot process related to the selection of sources and data extraction, which will help us in developing this guide.

I addition to the mentioned points, reviewer 1 had two minor comments, that we have answered: 

 1. This manuscript has some minor grammar and spelling errors that should be fixed (e.g., the research question states “Wath knowledge…”). 2. The author report funding but their PRISMA checklist says funding is “not applicable.” 

Answer: Thanks for making us aware of this. We have fixed those mistakes. 

Thank you for considering this significantly revised version, we hope you will find it suitable for publication.

Kind regards, 

Bjarnhild Samland, Tone Larsen and Lillian Pedersen

March 14, 2024

---

## [Editor Report · Decision Letter 1]

11 Jul 2024

Knowledge about research and facilitation of co-creation with children. Protocol for the article “Scoping review of research about co-creation with children”

PONE-D-23-36267R1

Dear Dr. Samland,

We’re pleased to inform you that your manuscript has been judged scientifically suitable for publication and will be formally accepted for publication once it meets all outstanding technical requirements.

Kind regards,

Sherief Ghozy, M.D.

Academic Editor

PLOS ONE

---

## [Editor Report · Acceptance letter]

1 Aug 2024

PONE-D-23-36267R1 

PLOS ONE

Dear Dr. Samland, 

I'm pleased to inform you that your manuscript has been deemed suitable for publication in PLOS ONE. Congratulations! Your manuscript is now being handed over to our production team.

Kind regards, 

on behalf of

Dr. Sherief Ghozy 

Academic Editor

PLOS ONE